# The Preparation of Robust Gully-like Surface of Stainless Steel Fiber-Bonded TFPA–TTA–COF with Nano Pores for Solid-Phase Microextraction of Phenolic Compounds in Water

**DOI:** 10.3390/nano15050354

**Published:** 2025-02-24

**Authors:** Wanqian Wei, Yu Shi, Keqing Zhang, Baohui Li

**Affiliations:** 1Department of Environmental Science and Engineering, North China Electric Power University, Baoding 071003, China; wwqtotoro@163.com (W.W.);; 2Hebei Key Laboratory of Power Plant Flue Gas Multi-Pollutants Control, Baoding 071003, China

**Keywords:** solid-phase microextraction, gas chromatography, covalent organic frameworks, phenolic compounds, gully-like surface

## Abstract

In this paper, a novel robust TFPA–TTA–COF coating with nano pores was grafted to the gully-like surface of stainless steel fibers (GS-SSF). The GS-SSF were prepared using a two-step electrochemical etching method, and the covalent organic framework (COF) TFPA–TTA–COF coating was chemically bonded to the gully-like surface via in situ growth. The prepared metal fibers were applied as the headspace solid-phase microextraction (HS-SPME) fibers and combined with gas chromatography (GC) to develop a detection method for phenolic compounds (PCs) in water. The developed method gave the limits of detection (S/N = 3) from 0.07 µg·L^−1^ to 0.52 µg·L^−1^ with enrichment factors from 243 to 2405. The relative standard deviations for inter-day study (n = 5) and fiber-to-fiber were from 3.94% to 8.89% and 2.17% to 8.05%, respectively. The prepared fiber could stand at least 180 cycles without remarkable loss of extraction efficiency. The developed method was successfully employed for the determination of trace PCs in environmental water with recoveries from 84.76% to 124.84%.

## 1. Introduction

Phenolic compounds (PCs) are a class of aromatic compounds in which hydrogen atoms on the benzene ring are replaced by hydroxyl groups and other substituents. PCs are widely used as industrial materials, including preservatives [1], disinfectants [2], antioxidants [3], dye intermediates [4], flame retardants [5], etc. Most PCs in the environment are mainly derived from the industrial wastewater of the chemical [6], pharmaceutical [7], and petrochemical industries [8], among others. Once PCs are discharged into the aquatic environment, they could be harmful to aquatic creatures and damage the water environment even at low concentrations [9]. By invading the human body through the food chain, PCs could lead to cell inactivation and damage to the liver, kidneys, and vascular system of humans [10]. Due to their high toxicity, persistence, and widespread presence, the detection of PCs in wastewater is of great importance. Given the low concentration of PCs in water, it is necessary to enrich the PCs prior to chromatographic separation.

Solid-phase microextraction (SPME) is a highly efficient, fast, and green sample pretreatment technique that integrates extraction, concentration, injection, and desorption into one step, which was first introduced by Pawlinszyn and Arthur in 1990 [11]. SPME is widely applied in the fields of environmental monitoring [12,13], pharmaceutical analysis [14,15], pesticide residue detection [16,17], etc. Traditional SPME fibers are typically made of fragile quartz [18,19], which leads to the shortening of the service time of the SPME fibers. Therefore, all kinds of metal materials were employed as the fiber substrate, such as titanium fibers [20,21], copper fibers [22,23], stainless steel fibers (SSF) [24,25], etc. In order to expand the application fields of metal fibers, various coatings were exhibited. Conventional commercial fiber coatings, such as polyacrylate [26,27], poly(dimethyl siloxane) (PDMS) [28,29], poly(dimethyl siloxane)/divinylbenzene (PDMS/DVB) [30,31], etc., often suffered from poor selectivity and low enrichment efficiency. To overcome these disadvantages, new coating materials have been increasingly investigated, including polymeric ionic liquids [32,33], graphene [34,35], metal–organic frameworks [36,37], molecularly imprinted polymers [38,39], etc.

Covalent organic frameworks (COFs), which are a class of porous materials composed only via covalent bonds, have the features of high surface area, uniform porosity, low density, and flexible design [40]. COFs have been widely used in various fields, including adsorption [41,42], catalysis [43,44], sensing [45,46], etc. Currently, various COFs are employed in the pretreatment field as SPME coating for different analytes [47,48,49,50].

In fact, owing to the repeated friction between the fiber and the inner wall of the syringe during the extraction process, the coating of the fiber is easily exfoliated, which could decrease the stability of the metal fibers and shorten the service time of the fibers. To resolve these problems, Li and co-workers [51] tried to graft COF-TpBD in the arrayed nanopores of SSF, which was fabricated with an electrochemical etching method.

In this paper, we present a more robust TFPA–TTA–COF coating bound to the gully-like surface of SSF (GS-SSF), which was prepared using a two-step electrochemical etching method. The TFPA–TTA–COF coating was grafted to the gully-like surface via in situ growth. The prepared metal fibers were employed to enrich and detect PCs in water by combining gas chromatography (GC) with flame ionization detection (FID), and excellent stability and high reproducibility were exhibited.

## 2. Experimental

### 2.1. Chemicals and Reagents

All chemicals and reagents were at least of analytical grade without further purification. Methanol (99.9%), anhydrous ethanol (≥99.7%), ethylene glycol (99.0%), ammonium fluoride (≥96.0%), sodium chloride (≥99.5%), acetic acid (36%), and N,N-dimethylformamide (DMF, ≥99.5%) were purchased from Tianjin Kermel Chemical Reagent Co., Ltd. (Tianjin, China). Acetone (≥99.0%), benzene (≥99.5%), and toluene (99.5%) were obtained from Tianjin Damao Chemical Reagent Factory (Tianjin, China). 4,4′,4″-(1,3,5-triazine-2,4,6-triyl) triphenylamine (TTA), tris(4-formylphenyl) amine (TFPA), and 3-aminopropyltriethoxysilane (APTES) were purchased from Shanghai Dibai Chemical Co., Ltd. (Shanghai, China). Tetrahydrofuran (≥99.0%), m-cresol (3-MP, ≥99.0%), 2,6-dimethylphenol (2,6-DMP, ≥99.0%), o-nitrophenol (2-NP, ≥98.0%), 2,4-dimethylphenol (2,4-DMP, ≥98.0%), 2,4-dichlorophenol (2,4-DCP, ≥98.0%), octadecane (≥98.0%), eicosane (≥99.0%), aniline (≥99.5%), o-toluidine (≥99%), n-butanol (≥99.7%), and isopropanol (≥99.7%) were sourced from Aladdin Chemical Co., Ltd. (Shanghai, China). Perchloric acid (≥70.0%) was purchased from Tianjin Zhengcheng Chemical Products Co., Ltd. (Tianjin, China). N_2_ was supplied by Hanjiangxue Trading Co., Ltd. (Baoding, China). Ultrapure water collected from a Milli-Q integral system (Millipore China Co., Ltd., Shanghai, China) was used throughout the experiment.

### 2.2. Instrumentation

The GC 2014C gas chromatography with FID was purchased from Shimadzu Co. (Kyoto, Japan). The HP-5 chromatographic column (30 m × 0.32 mm ID × 0.5 µm) was obtained from Lanzhou Institute of Chemical Physics (Lanzhou, China). The DP800A series programmable linear DC power supply was sourced from Beijing Puyuanjingdian Technology Co., Ltd. (Beijing, China). The 5 µL syringes were from Shanghai Gaoge Industrial and Trade Co., Ltd. (Shanghai, China). Commercial SPME fibers PDMS (30 µm) and DVB/CAR/PDMS (50/30 µm) were procured from Sigma-Aldrich Trading Co., Ltd. (Shanghai, China).

The Fourier transform infrared (FT-IR) spectra were measured on a Tensor II (Bruker AXS GmbH, Karlsruhe, Germany). The thermogravimetric analysis (TGA) was performed on a TGA 4000 thermal gravimetric analyzer (Rigaku, Tokyo, Japan). The scanning electron microscope (SEM) images were recorded on a Hitachi S4800 (Hitachi, Tokyo, Japan). The X-ray diffraction (XRD) data were obtained using a D8 Advance diffractometer (Bruker AXS GmbH, Germany).

### 2.3. Chromatographic Condition

The high-purity N_2_ was used as the carrier gas at a flow rate of 30 mL∙min^−1^, and the H_2_ and the air flow rates were adjusted to 40 mL∙min^−1^ and 400 mL∙min^−1^, respectively. The injector temperature was maintained at 260 °C, and the detector temperature at 280 °C. The column temperature program started at 50 °C, was kept for 2 min, and then increased to 150 °C at 10 °C·min^−1^, followed by 20 °C·min^−1^ to 220 °C, which was kept for 2 min. All injections were performed in splitless mode.

### 2.4. The Fabrication of the COF

TFPA (198 mg) and TTA (213 mg) were added into the liner of the hydrothermal synthesis reactor containing 35 mL of DMF, followed by the addition of 0.5 mL acetic acid (6 mol∙L^−1^) [52]. The liner was transferred to the hydrothermal synthesis reactor with magnetic stirring at the agitation speed of 250 rpm. The temperature of the hydrothermal synthesis reactor was set to increase from 30 °C to 90 °C at a rate of 2 °C·min^−1^ and maintained for 48 h. After the reaction finished, the yellow solid TFPA–TTA–COF was filtered out, and then washed with acetone and anhydrous ethanol until the filtrate became colorless and transparent. At last, the TFPA–TTA–COF was dried under vacuum at 60 °C for 12 h.

### 2.5. The Fabrication of the Etched SSF

The SSF were physically polished with sandpaper and ultrasonically cleaned with acetone, anhydrous ethanol, and ultrapure water for 10 min, respectively. Then, the SSF was put into an electrolytic bath of the DP800A series programmable linear DC power, and a two-step electrochemical etching method was used to etch the SSF. A commercial graphite tube was used as the cathode and the SSF as the anode, which was put into the center line of the graphite tubes.

A voltage of 25 V was chosen for the 50 s etching process, and the electrolyte was a mixture of 10% perchloric acid and 90% ethylene glycol. Secondly, a voltage of 50 V was chosen to be applied for 5 min, and the electrolyte was a mixture of 0.1 mol·L^−1^ ammonium fluoride and 0.1 mol·L^−1^ ultrapure water mixed with ethylene glycol. Finally, the fibers were rinsed with ultrapure water, and the GS-SSF was obtained.

### 2.6. The Fabrication of the TFPA–TTA–COF Coated on GS-SSF

The schematic illustration of the fabrication processes of the SPME fibers is shown in Figure 1. Firstly, the dried GS-SSF were immersed in a mixed solution of APTES and anhydrous ethanol (V:V, 1:4), and the reaction was carried out in an oil bath at 50 °C for 8 h. Secondly, the fibers were dried under nitrogen to obtain amino-functionalized stainless steel fibers (NH_2_-GS-SSF). Then, the NH_2_-GS-SSF were immersed in a mixed solution of TFPA and TTA and reacted in the hydrothermal synthesis reactor with magnetic stirring for 48 h. After, the TFPA–TTA–COF was chemically bonded to the stainless steel fibers (TFPA–GS-SSF). At last, the TFPA–GS-SSF were rinsed with acetone and anhydrous ethanol to remove the unreacted ligands and vacuum-dried at 60 °C for 12 h.

### 2.7. SPME Procedure

All extractions were conducted using headspace solid-phase microextraction (HS-SPME) mode. Before the SPME procedure, the prepared TFPA–GS-SSF were conditioned in the GC injection port at 260 °C for 15 min. Then, the TFPA–GS-SSF was exposed to the headspace of a 20 mL headspace vial with 10 mL working solution for 30 min. After, the TFPA–GS-SSF were removed from the headspace vial and immediately transferred to the GC injection port for thermal desorption at 260 °C for 2 min. During the experiment, a thermostatic water bath was used to control the extraction temperature, and a magnetic stirrer was used to regulate the extraction agitation rate.

## 3. Results and Discussion

### 3.1. The Characterization of the TFPA–TTA–COF

The FT-IR spectrum comparison of TFPA–TTA–COF with TTA and TFPA (Figure 2a) showed that the characteristic band of N–H from TTA at 3207 cm^−1^, 3321 cm^−1^, and 3461 cm^−1^, as well as the characteristic band of C–H from TFPA at 2733 cm^−1^ and 2812 cm^−1^, disappeared in the FT-IR spectra of TFPA–TTA–COF. In addition, a new characteristic band of C=N from TFPA–TTA–COF appeared at 1622 cm^−1^. The results confirmed the successful synthesis of the TFPA–TTA–COF. The TGA analysis curve showed good stability of TFPA–TTA–COF between 200 °C and 400 °C (Figure 2b), which met the requirements of GC analysis. The morphologies of the prepared TFPA–TTA–COF were characterized by SEM (Figure 2c) and the nano-sized pore was observed. The N₂ adsorption isotherms obtained by the Barrett–Joyner–Hallenda method (Figure 2d), and the specific surface area, pore volume, and average pore size of TFPA–TTA–COF were 44.359 m^2^∙g^−1^, 0.055 m^3^∙g^−1^, and 4.922 nm, respectively.

### 3.2. The Characterization of the SPME Fibers

In comparison with the FT-IR (Figure 3a) spectra and XRD pattern (Figure 3b) of TFPA–TTA–COF, the ones of the TFPA–TTA–COF coatings scraped from the prepared fiber did not alter remarkably. The results revealed that the characteristic bands and crystallinity of TFPA–TTA–COF fabricated with two approaches were identical, and indicated that the TFPA–TTA–COF was successfully synthesized and grafted onto the GS-SSF surface.

The SEM images showed that the surface of SSF formed a rough, gully-like structure (Figure 4a–c), which provided more binding sites. The SEM images also revealed the presence of TFPA–TTA–COF as a homogenous coating on the surface of SSF (Figure 4d–f). The coating provided more adsorption sites for analytes and improved the enrichment capacity of the TFPA–GS-SSF.

### 3.3. The Optimization of the Extraction and Desorption Conditions

The potential parameters influencing the extraction efficiency, such as extraction time, extraction temperature, agitation speed, ionic strength, desorption temperature, and desorption time were optimized.

The effect of extraction time on the peak areas of the PCs was examined from 15 min to 40 min (Figure 5a). The extraction of 3-MP and 2,4-DMP reached equilibrium within 25 min, while the extraction of 2,6-DMP, 2-NP, and 2,4-DCP reached equilibrium within 30 min. Therefore, 30 min was set as the optimized extraction time for subsequent experiments.

The effect of extraction temperature was studied from 30 °C to 70 °C (Figure 5b). The peak areas of the PCs increased steadily as the extraction temperature increased from 30 °C to 50 °C. The further increase of the extraction temperature from 50 °C to 70 °C led to the reduction of the peak areas. On one side, the high extraction temperature helps to accelerate the mass transfer of PCs from the solution to the headspace and increase the concentration of PCs in the headspace. On the other side, the excessively high extraction temperature hindered the adsorption process since it is an exothermic adsorption. Therefore, 50 °C was chosen as the optimized extraction temperature for subsequent experiments.

The effect of desorption time was investigated from 0.5 min to 2 min (Figure 5c). The peak areas of the PCs increased from 0.5 min to 2 min. The further increase of the extraction temperature resulted in the broadening of peaks, so 2 min was chosen as the upper limit for desorption time. While the desorption time was 2 min, the peak areas of the PCs gradually increased as the desorption temperature increased from 230 °C to 260 °C. With the further improvement of the desorption temperature, the peak areas of the PCs remained unchanged (Figure 5d). Therefore, the desorption parameters were set at 260 °C for 2 min in the subsequent experiments.

The effect of agitation speed was examined from 400 rpm to 800 rpm (Figure 5e). The peak areas of the PCs improved with the varying of the agitation speed from 400 rpm to 600 rpm. The further increase of the agitation speed from 600 rpm to 800 rpm led to the fluctuation of the peak areas because of the unstable solution. Therefore, 600 rpm was chosen as the optimized agitation speed for subsequent experiments.

The effect of ionic strength was tested, and the NaCl concentration (W/V) varied from 0% to 35% (Figure 5f). In general, the high ionic strength can cause a salt-out effect and decrease the solubility of organic compounds in an aqueous solution, which can accelerate the release of the analytes from the aqueous solution. The peak areas of PCs continually increased with the salt concentration from 0% to 35%. Seeing that the solubility of NaCl in water is 36% at room temperature, the higher NaCl concentration could not be investigated. Therefore, 35% NaCl concentration was selected as the optimized ionic strength for subsequent experiments.

In consideration of the above discussion, the optimized HS-SPME conditions for PCs were determined, including the 30 min extraction time, the 50 °C extraction temperature, the 2 min desorption time, the 260 °C desorption temperature, the 600 rpm agitation speed, and the 35% (NaCl, W/V) ionic strength. Under the optimized conditions, the chromatogram of PCs in water demonstrated a satisfactory separation (Figure 6).

### 3.4. The Durability and Selectivity of the Prepared SPME Fiber

Generally, for the whole SPME experiment, including the pre-experiment, the parameter optimization, the figures of merit, and the sample analysis, there should be at least 150 extraction cycles. So, the SPME fiber needed to withstand at least 150 extraction cycles to improve reproducibility. It can be seen after 180 extraction cycles that the extraction efficiency of the prepared TFPA–GS-SSF did not decrease remarkably, and the error of the peak area did not vary much when comparing 30 extraction cycles at the same time (Figure 7a), which met the experimental requirements completely. As a matter of fact, after 12 months of storage, the performance of the prepared fiber varied slightly. All the results improved the stability of the prepared fibers.

The adsorption selectivity of the prepared TFPA–GS-SSF was tested, with octadecane, eicosane, benzene, toluene, aniline, o-toluidine, n-butanol, and isopropanol as the analytes (Figure 7b). The results demonstrated that TFPA–GS-SSF not only offered excellent extraction performance for PCs but also exhibited impressive extraction selectivity.

### 3.5. The Comparison of the Different SPME Fibers

To confirm that the adsorption ability of the prepared TFPA–GS-SSF for PCs came from the TFPA–TTA–COF rather than the amino group or the bare stainless steel surface, a chromatogram comparison of the prepared TFPA–GS-SSF with the aminated stainless steel fiber and the bare one was given (Figure 8a). It can be seen that a positive result was obtained. It was attributed to the large surface area of TFPA–TTA–COF, which provided ample adsorption sites for PCs. Two kinds of commercial fibers were also employed for the extraction of PCs to examine the extraction selectivity and extraction efficiency (Figure 8b). The enrichment factors (EFs) were used for quantitative analysis, and the EFs were defined as the ratio of the peak area after SPME to that obtained by direct injection of 1 µL of a standard solution. The TFPA–GS-SSF gave much larger EFs than the commercial fibers, and the results revealed the great extraction potential of TFPA–GS-SSF for PCs.

The large-scale production of TFPA–GS-SSF is feasible since the main procedure involves in situ hydrothermal synthesis, whose reaction conditions can be controlled by the temperature and the rotary speed of the hydrothermal kettle. While TFPA–TTA–COF was replaced with the other COFs with different polar groups and cavities, more SPME fibers with specific adsorption can be obtained since we may select the right COFs depending on the molecular size and the hydrophobicity of the analytes. So, we firmly believe that the scalability of the fibers has potential.

### 3.6. The Analytical Figures of Merit

The analytical performances of the developed method are summarized in Table 1. The linear range was 5–1000 μg∙L^−1^ for 3-MP and 2-NP, 2–400 μg∙L^−1^ for 2,4-DMP and 2,4-DCP, and 1–200 μg∙L^−1^ for 2,6-DMP. The linear correlation coefficients (R^2^) ranged from 0.9938 to 0.9984. The limits of detection (LODs, S/N = 3) varied from 0.07 μg∙L^−1^ to 0.52 μg∙L^−1^. The relative standard deviations (RSDs) for the inter-day study (n = 5) were in the range of 3.94% to 8.89%, and the RSDs of fiber-to-fiber for three parallel prepared fibers were in the range of 2.17% to 8.05%. The EFs for the PCs were from 243 to 2405. The results showed that the method had good repeatability and stability.

### 3.7. The Analysis of the PCs in Environmental Water

The developed method was employed for the SPME of the PCs in three environmental waters, including local lake water, Caohe River water, and tap water (Table 2). The recoveries ranged from 84.76% to 124.84%, obtained by spiking the standard PC solution in the environmental water, corresponding to spiking 3-MP and 2-NP at 50 µg·L^−1^, 2,4-DMP and 2,4-DCP at 20 µg·L^−1^, and 2,6-DMP at 10 µg·L^−1^ in the working solution. The results showed that the method was feasible for the determination of trace PCs in environmental water. The reason that the recoveries exceed 100% may be attributed to the complexity of the water sample since the analytes could be adsorbed to the surface of the microparticles in water. To make the recoveries more reasonable, in further experiments, an appropriate amount of organic solution could be added to the water sample to release the analytes. Furthermore, internal standards or blank corrections could be carried out to further validate the results.

## 4. Conclusions

In summary, we prepared a robust TFPA–TTA–COF coating, which was grafted to the gully-like surface of stainless steel fibers. The metal fibers were prepared using a two-step electrochemical etching method and the TFPA–TTA–COF coating was grafted to the gully-like surface via in situ growth. The prepared fibers were employed to determine trace PCs in the water by combining them with GC-FID. The developed method exhibited the limits of detection from 0.07 µg·L^−1^ to 0.52 µg·L^−1^ with enrichment factors from 243 to 2405, linear ranges from 5 to 1000 μg∙L^−1^, and linear correlation coefficients from 0.9938 to 0.9984, which was successfully employed for the determination of trace PCs in environmental water. The prepared fibers demonstrated good thermal stability and long service time. The fabrication method can be used to prepare more coatings of metal fibers to obtain long service times for SPME fiber.

## Figures and Tables

**Figure 1 nanomaterials-15-00354-f001:**
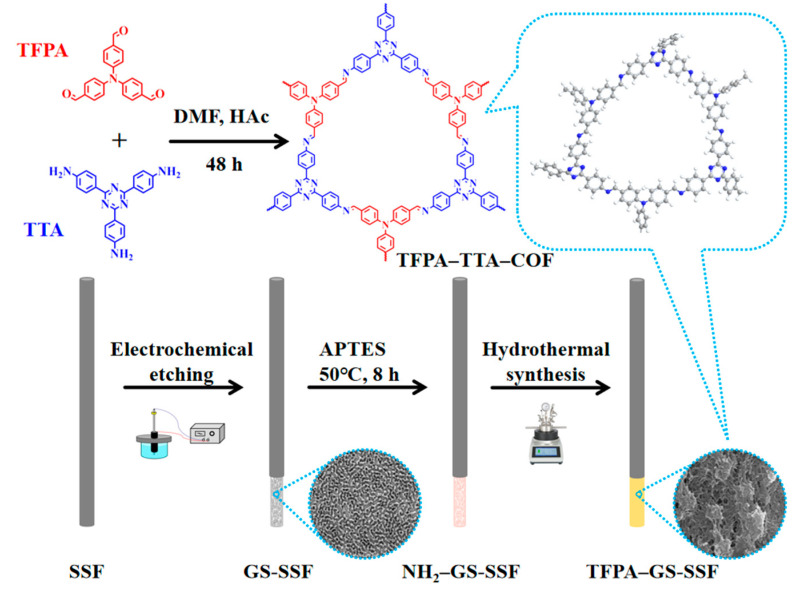
The schematic illustration of the fabrication processes of TFPA–GS-SSF.

**Figure 2 nanomaterials-15-00354-f002:**
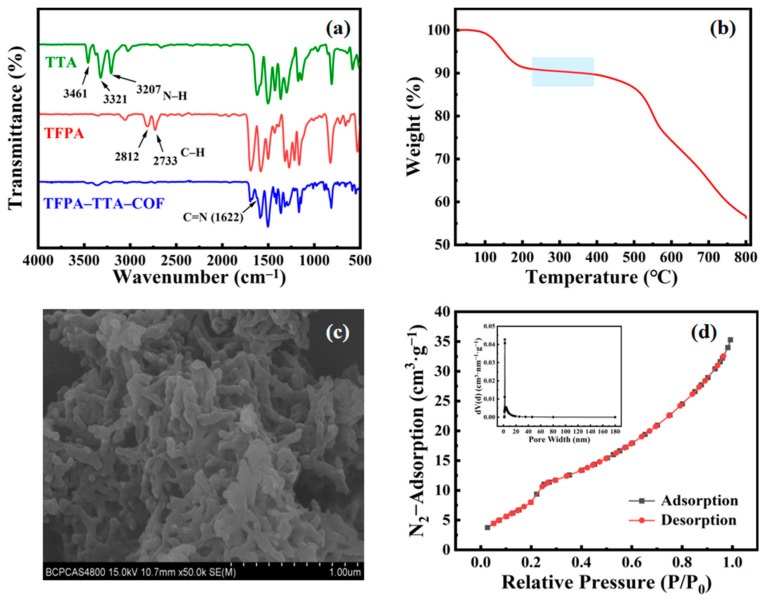
The characterization of TFPA–TTA–COF: (**a**) FT-IR comparison spectra; (**b**) TGA analysis curve; (**c**) SEM image at 50,000× magnification; (**d**) N_2_ adsorption isotherm and pore size distribution.

**Figure 3 nanomaterials-15-00354-f003:**
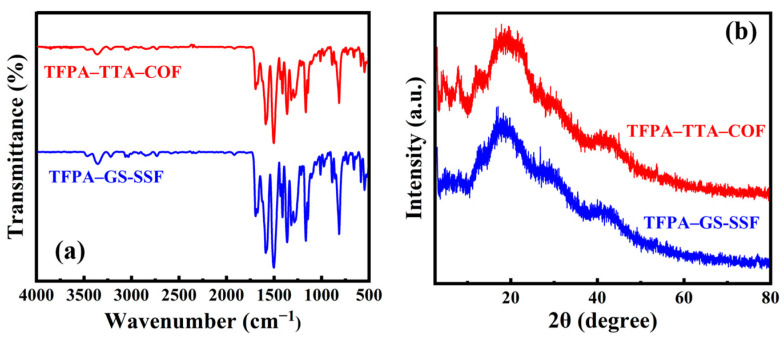
The characterization of SPME fibers: (**a**) FT-IR spectral comparison; (**b**) XRD spectral comparison.

**Figure 4 nanomaterials-15-00354-f004:**
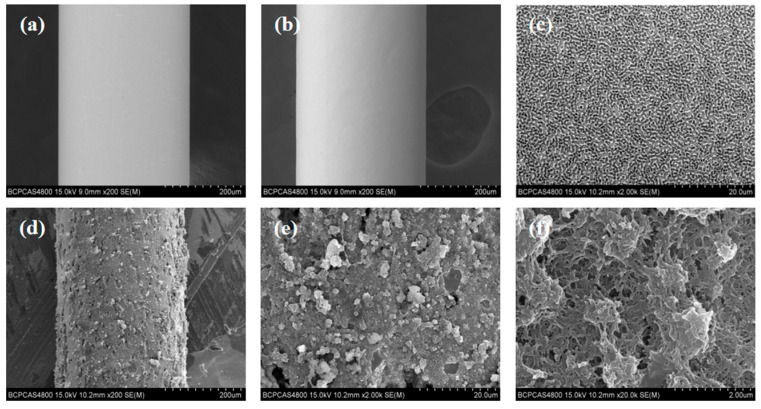
(**a**) SEM image of SSF at 200× magnification. SEM image of GS-SSF at various magnifications: (**b**) 200×; (**c**) 2000×. SEM image of TFPA–GS-SSF at various magnifications: (**d**) 200×; (**e**) 2000×; (**f**) 20,000×.

**Figure 5 nanomaterials-15-00354-f005:**
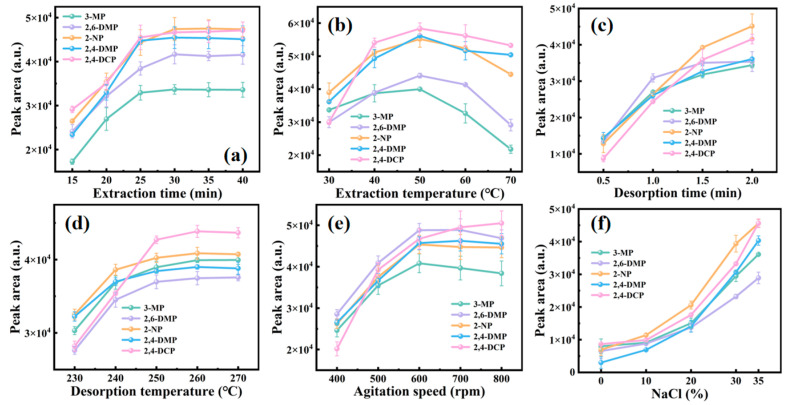
The effects of extraction parameters on the extraction and desorption efficiency of PAEs. (**a**) Extraction time (conditions: extraction temperature, 50 °C; desorption time, 2 min; desorption temperature, 260 °C; NaCl concentration, 35%; agitation speed, 600 rpm). (**b**) Extraction temperature (conditions: extraction time, 30 min; desorption time, 2 min; desorption temperature, 260 °C; NaCl concentration, 35%; agitation speed, 600 rpm). (**c**) Desorption time (conditions: extraction time, 30 min; extraction temperature, 50 °C; desorption temperature, 260 °C; NaCl Concentration, 35%; agitation speed, 600 rpm). (**d**) Desorption temperature (conditions: extraction time, 30 min; extraction temperature, 50 °C; desorption time, 2 min; NaCl concentration, 35%; agitation speed, 600 rpm). (**e**) Agitation speed (conditions: extraction time, 30 min; extraction temperature, 50 °C; desorption time, 2 min; desorption temperature, 260 °C; NaCl concentration, 35%). (**f**) NaCl concentration (conditions: extraction time, 30 min; extraction temperature, 50 °C; desorption time, 2 min; desorption temperature, 260 °C; agitation speed, 600 rpm).

**Figure 6 nanomaterials-15-00354-f006:**
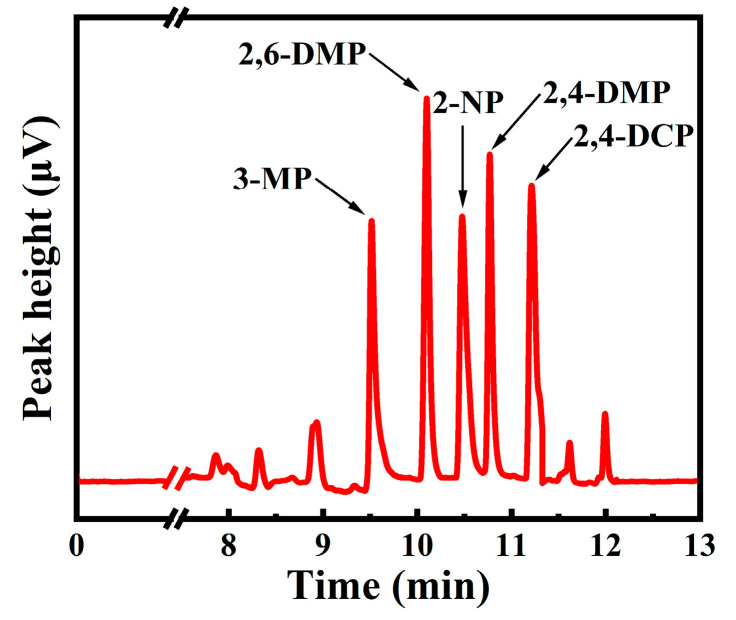
The chromatogram of five kinds of PCs in optimized conditions. (A mixture of 3-MP and 2-NP at 50 µg·L^−1^, 2,4-DMP and 2,4-DCP at 20 µg·L^−1^, and 2,6-DMP at 10 µg·L^−1^. Extraction conditions: extraction time of 30 min, extraction temperature of 50 °C, agitation speed of 600 rpm, and ionic strength of 35%. Desorption conditions: desorption time of 2 min and desorption temperature of 260 °C).

**Figure 7 nanomaterials-15-00354-f007:**
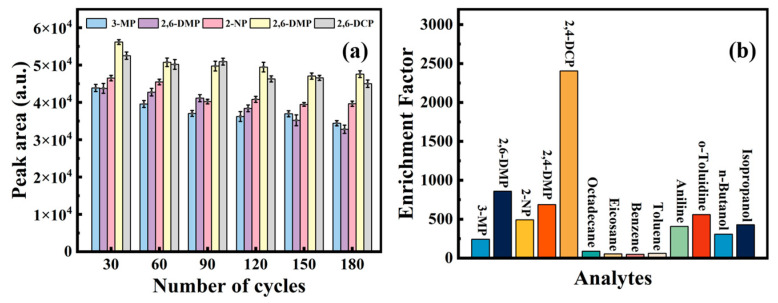
TFPA–GS-SSF. (**a**) The effect of usage frequency on extraction efficiency. (**b**) The comparison of different analytes.

**Figure 8 nanomaterials-15-00354-f008:**
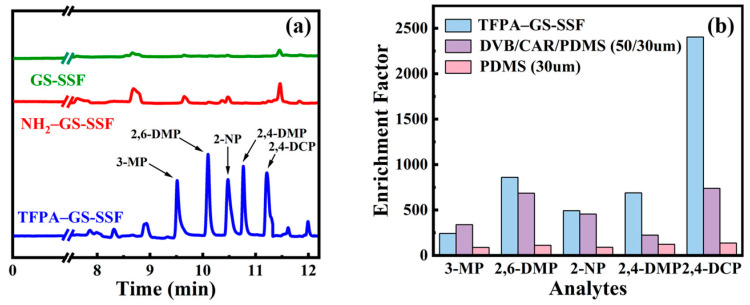
The comparison of extraction efficiency of different SPME fibers under optimized conditions. (**a**) GS-SSF, NH_2_-GS-SSF, and TFPA–GS-SSF; (**b**) TFPA–GS-SSF, commercial fiber DVB/CAR/PDMS, and commercial fiber PDMS.

**Table 1 nanomaterials-15-00354-t001:** Analytical performance of the TFPA–GS-SSF for the SPME of PCs.

Analytes	Linear Range (µg L^−1^)	R^2^	LODs (μg∙L^−1^)	RSDs		EFs *
				Inter-Day (%, n = 5)	Fiber-to-Fiber (%, n = 3)	
3-MP	5–1000	0.9938	0.17	5.87	3.68	243
2,6-DMP	1–200	0.9953	0.15	7.11	5.20	861
2-NP	5–1000	0.9984	0.52	8.89	8.05	493
2,4-DMP	2–400	0.9963	0.33	3.94	7.46	689
2,4-DCP	2–400	0.9966	0.07	4.98	2.14	2405

* The EF for this experiment is calculated as EF = (S_SPME_ × 100)/S_DI_ (S_DI_ refers to the peak area obtained by direct injection of 1 µL of a mixture containing 3-MP and 2-NP at 5000 µg·L^−1^, 2,4-DMP and 2,4-DCP at 2000 µg·L^−1^, and 2,6-DMP at 1000 µg·L^−1^; S_SPME_ refers to the peak area measured using the SPME-GC method from a 20 mL mixture containing 3-MP and 2-NP at 50 µg·L^−1^, 2,4-DMP and 2,4-DCP at 20 µg·L^−1^, and 2,6-DMP at 10 µg·L^−1^).

**Table 2 nanomaterials-15-00354-t002:** Analytical results of PCs in the environmental water.

Analytes	Lake Water		Caohe River Water		Tap Water	
	Found (μg∙L^−1^)	Recoveries (%)	Found (μg∙L^−1^)	Recoveries (%)	Found (μg∙L^−1^)	Recoveries (%)
3-MP	ND *	100.46	ND	100.56	ND	90.38
2,6-DMP	ND	111.65	ND	110.85	ND	100.34
2-NP	ND	100.25	ND	106.61	ND	84.76
2,4-DMP	ND	91.38	ND	108.55	ND	113.19
2,4-DCP	ND	122.06	ND	102.32	ND	124.84

* ND = not detected.

## Data Availability

Data are contained within the article.

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
