# Peer review of "The Preparation of Robust Gully-like Surface of Stainless Steel Fiber-Bonded TFPA–TTA–COF with Nano Pores for Solid-Phase Microextraction of Phenolic Compounds in Water"

_nanomaterials, 2025, doi:10.3390/nano15050354_

Round 1

Reviewer 1 Report

Comments and Suggestions for Authors

After reviewing the manuscript "The Preparation of Robust Gully-like Surface of Stainless-Steel Fiber Bonded TFPA-TTA-COF with Nano Pore for Solid-Phase Microextraction of Phenolic Compounds in Water", I have the following comments:

1. The paper presents a novel, robust covalent organic framework-coated stainless steel fiber for solid-phase microextraction, demonstrating enhanced selectivity and durability in detecting trace phenolic compounds in water.

2. The study claims reproducibility and durability with 180 extraction cycles, but no clear standard deviation data or statistical backing is given beyond a general claim. This could raise concerns about the actual robustness in industrial applications.

3. The comparison of enrichment factors (Figure 8) highlights that the new fibers perform better than commercial PDMS/DVB fibers. However, the study doesn’t dive into the practical side of things, like how the operational costs or maintenance needs compare, something that would be key for real-world use.

4. While the fibers demonstrate impressive selectivity for phenolic compounds (PCs), other competing analytes like heavy metals, pesticides, or halogenated compounds were not evaluated. This limits the general applicability of the method for diverse water samples.

5. The recovery values (84.76% to 124.84%) are quite broad, and values over 100% suggest potential issues with matrix effects, calibration errors, or interference in environmental samples.

6. Thermal and mechanical stability tests (e.g., after long-term storage) are missing. These would provide insights into degradation patterns that affect reusability.

7. Figure 3: There is no quantitative comparison of peak intensities or crystallinity measurements. Providing data like peak area ratios or crystallite size estimations would make the validation stronger.

8. The manuscript’s language is generally clear, but it contains several typos, grammar issues, and awkward phrases that could affect readability. A thorough proofreading pass is recommended to improve fluency and consistency.

Comments on the Quality of English Language

The English is generally clear and gets the message across, but there are some grammar issues, typos, and awkward phrases that could make it hard to follow in places. With a bit of polishing and careful proofreading, the writing can be much smoother and easier to read.

Author Response

Comments 1: The paper presents a novel, robust covalent organic framework-coated stainless steel fiber for solid-phase microextraction, demonstrating enhanced selectivity and durability in detecting trace phenolic compounds in water.

Response 1: Thank you for reviewing our paper, and I will provide a point-by-point response to your comments.

Comments 2: The study claims reproducibility and durability with 180 extraction cycles, but no clear standard deviation data or statistical backing is given beyond a general claim. This could raise concerns about the actual robustness in industrial applications.

Response 2: Thank you for pointing this out. I agree with this comment. We added absolute error data in the figure 7(a), and the relative paragraph was revised in the manuscript. (See page 8, line 256).

Comments 3: The comparison of enrichment factors (Figure 8) highlights that the new fibers perform better than commercial PDMS/DVB fibers. However, the study doesn’t dive into the practical side of things, like how the operational costs or maintenance needs compare, something that would be key for real-world use.

Response 3: Thank you for pointing this out. In fact, the purpose of our research is to grope for high extraction efficiency fiber for PCs. So, the practical side of the fiber was not be discussed. In future if the prepared fiber could be mass-produced, we will dive into the practical side of the prepared fiber.

Comments 4: While the fibers demonstrate impressive selectivity for phenolic compounds (PCs), other competing analytes like heavy metals, pesticides, or halogenated compounds were not evaluated. This limits the general applicability of the method for diverse water samples.

Response 4: Thank you for pointing this out. Since the selective adsorption of COFs relies on the polarity, cavity diameter and π-π conjugation, we chosen alkanes, benzenes, anilines and alcohols as the competing analytes. Owing to the poor adsorption efficiency of heavy metals, they were removed from the competing analytes list before the extraction experiment was launch. Since the FID of GC for the compounds containing sulfur or phosphor is insensitive, Pesticides or halogenated compounds were not selected. Your review is valuable and we will evaluate more competing analytes in the further experiments.

Comments 5: The recovery values (84.76% to 124.84%) are quite broad, and values over 100% suggest potential issues with matrix effects, calibration errors, or interference in environmental samples.

Response 5: Thank you for pointing this out. Although the range of recovery rate values in the paper is wide, they are acceptable data (See DOI: 10.1016/j.cej.2019.03.148 and DOI: 10.1016/j.jhazmat.2023.131382). In fact, for all the SPME experiments the range of recovery rate values would broaden unavoidably when the developed method was use to detect real sample. We will further standardize the experimental operation to exclude the influence of various factors and shorten the range of recovery rate values.

Comments 6: Thermal and mechanical stability tests (e.g., after long-term storage) are missing. These would provide insights into degradation patterns that affect reusability.

Response 6: Thank you for pointing this out. We performed a thermogravimetric analysis of the COF material (See Fig. 2b), and the results showed that it had good thermal stability at GC run temperature. To test the mechanical stability of the prepared fibers, 180 cycle experiment was carried out. After 180 extraction cycles, the extraction efficiency of the prepared fiber did not decrease remarkably. The results showed the good mechanical stability of the prepared fiber. In fact, our experiment lasted at least 12 months. So, the long-term mechanical stability was added and the relative paragraph was revised in the manuscript. (See page 8, line 258).

Comments 7: Figure 3: There is no quantitative comparison of peak intensities or crystallinity measurements. Providing data like peak area ratios or crystallite size estimations would make the validation stronger.

Response 7: Thank you for pointing this out. The information you have provided is valuable. However, in our manuscript the XRD characterization spectra of the fabricated COF materials compared with the COF coating. In section 2.4, reference 52 proved the successful fabrication of the COF. It is enough to prove the successful synthesis of the COF coating.

Comments 8: The manuscript’s language is generally clear, but it contains several typos, grammar issues, and awkward phrases that could affect readability. A thorough proofreading pass is recommended to improve fluency and consistency.

Response 8: Thank you for pointing this out. I agree with this comment. Therefore, we did a thorough proofreading of the paper to improve the fluency and consistency of the manuscript. In the revised manuscript the alteration can be found in the text marked in red.

Reviewer 2 Report

Comments and Suggestions for Authors

The authors submitted their paper titled “The Preparation of Robust Gully-like Surface of Stainless-Steel 2 Fiber Bonded TFPA-TTA-COF with Nano Pore for Solid-Phase 3 Microextraction of Phenolic Compounds in Water” for publication in the Journal of Nanomaterials.They prepared metal fibers. These fibers were applied as the headspace solid-phase micro extraction (HS-SPME). Fibers combined with gas chromatography (GC) to develop a detection method for phenolic compounds (PCs) in water. The prepared TFPA-GS-SSF demonstrated good thermal stability and long service time.

The paper contains an original scientific contribution. The procedure is precisely described and all results are presented in a comprehensible manner. The figures and tables are clear and technically well presented.

The conclusion of the paper should be more comprehensive. It should be expanded and include more data that illustrate the final result. In addition, the conclusion is burdened with a large number of abbreviations, which makes it difficult for the reader to understand what the fundamental contribution consists of.

Author Response

Comments 1: The conclusion of the paper should be more comprehensive. It should be expanded and include more data that illustrate the final result. In addition, the conclusion is burdened with a large number of abbreviations, which makes it difficult for the reader to understand what the fundamental contribution consists of.

Response 1: Thank you for pointing this out. I agree with this comment. Therefore, we revise the conclusion, and make it more comprehensive. In the revised manuscript the alteration can be found - page 10 and line 308.

Round 2

Reviewer 1 Report

Comments and Suggestions for Authors

The authors have made significant improvements to the manuscript by addressing key concerns related to fiber durability, selectivity, and long-term stability. The inclusion of statistical data strengthens the reproducibility claims, and the revised explanations improve clarity. However, a brief discussion on the real-world feasibility of the fiber, such as its manufacturability, operational costs, or potential scalability, would enhance the study’s practical relevance. Additionally, while the recovery rates fall within an acceptable range, values exceeding 100% may indicate matrix effects or calibration inconsistencies.

A short explanation of how these factors were controlled, such as through internal standards or blank corrections, would further validate the results. Overall, the manuscript is well-prepared for acceptance with minor refinements.

Author Response

Comments 1: A brief discussion on the real-world feasibility of the fiber, such as its manufacturability, operational costs, or potential scalability, would enhance the study’s practical relevance.

Response 1: Thank you for pointing this out. I agree with this comment. We added the discussion about the fabrication method of the real-world feasibility. The relative paragraph was revised in the manuscript (See page 9, line 284). While the potential scalability of the fibers were discussed in the manuscript (See page 9, line 286). Since the cost of the fibers were mainly from the monomer expense of the COFs and the amount per fiber is negligible, the operational costs was not discussed.

Comments 2: While the recovery rates fall within an acceptable range, values exceeding 100% may indicate matrix effects or calibration inconsistencies. A short explanation of how these factors were controlled, such as through internal standards or blank corrections, would further validate the results.

Response 2: Thank you for pointing this out. I agree with this comment. We added the explanation. The relative paragraph was revised in the manuscript (See page 10, line 313).